# Twelve Tips for Inclusive Practice in Healthcare Settings

**DOI:** 10.3390/ijerph20054657

**Published:** 2023-03-06

**Authors:** Brahmaputra Marjadi, Joanne Flavel, Kirsten Baker, Kristen Glenister, Melissa Morns, Mel Triantafyllou, Penelope Strauss, Brittany Wolff, Alexandra Marie Procter, Zelalem Mengesha, Scott Walsberger, Xiaoxi Qiao, Paul A. Gardiner

**Affiliations:** 1Diversity, Equity and Inclusion Special Interest Group, Public Health Association of Australia, Deakin, ACT 2600, Australia; 2Translational Health Research Institute, School of Medicine, Western Sydney University, Campbelltown, NSW 2560, Australia; 3Stretton Health Equity, Stretton Institute, School of Social Sciences, The University of Adelaide, Adelaide, SA 5005, Australia; 4Australian Research Consortium in Complementary and Integrative Medicine, Faculty of Health, University of Technology Sydney, Ultimo, NSW 2007, Australia; 5Department of Rural Health, The University of Melbourne, Wangaratta, VIC 3677, Australia; 6The Australian Centre for Public and Population Health Research, University of Technology Sydney, Ultimo, NSW 2007, Australia; 7Health Research Institute, University of Canberra, Bruce, ACT 2617, Australia; 8Telethon Kids Institute, The University of Western Australia, Nedlands, WA 6009, Australia; 9School of Psychological Sciences, The University of Western Australia, Crawley, WA 6009, Australia; 10School of Public Health, The University of Adelaide, Adelaide, SA 5005, Australia; 11Centre for Primary Health Care and Equity, The University of New South Wales, UNSW, Sydney, NSW 2025, Australia; 12ACON, Surry Hills, NSW 2010, Australia; 13School of Public Health, The University of Queensland, Herston, QLD 4006, Australia

**Keywords:** health service, diversity, equity, inclusion

## Abstract

This paper outlines practical tips for inclusive healthcare practice and service delivery, covering diversity aspects and intersectionality. A team with wide-ranging lived experiences from a national public health association’s diversity, equity, and inclusion group compiled the tips, which were reiteratively discussed and refined. The final twelve tips were selected for practical and broad applicability. The twelve chosen tips are: (a) beware of assumptions and stereotypes, (b) replace labels with appropriate terminology, (c) use inclusive language, (d) ensure inclusivity in physical space, (e) use inclusive signage, (f) ensure appropriate communication methods, (g) adopt a strength-based approach, (h) ensure inclusivity in research, (i) expand the scope of inclusive healthcare delivery, (j) advocate for inclusivity, (k) self-educate on diversity in all its forms, and (l) build individual and institutional commitments. The twelve tips are applicable across many aspects of diversity, providing a practical guide for all healthcare workers (HCWs) and students to improve practices. These tips guide healthcare facilities and HCWs in improving patient-centered care, especially for those who are often overlooked in mainstream service provision.

## 1. Introduction

Health is a basic human right, as outlined in the World Health Organization constitution, and the highest attainable state of health is a fundamental right of every human being [1]. Yet, marginalized groups experience poorer mental and physical health outcomes than the general population due to health inequities [2,3,4] largely driven by social determinants of health—the conditions in which people live throughout their lifespan [5]. There is a causal, vicious cycle between disadvantages due to social determinants of health and the poor health outcomes. Poor health status subjects people to additional disadvantages, for example, by limiting their ability to obtain education or engage in work. In turn, those who are disadvantaged often have limited access to healthcare services. One important factor for this lack of access is the notion that healthcare services may not be inclusive of the diverse populations who need them the most. Healthcare services which are not inclusive for diverse populations further contribute to these avoidable health inequities.

Awareness of health inequities has led to the development of inclusive, person-/patient-centered healthcare practices, which are practices tailored to individual identities, beliefs, and needs [6,7]. However, searches of the literature in 2021 and 2022 failed to identify inclusivity guides covering multiple diversity aspects and their intersections. Equally lacking was a set of practical steps which clinicians, frontline workers, managers, and health professional students could use as a quick guide for inclusive practice. To address this gap, the work for this paper was undertaken under the auspices of the Diversity, Equity, and Inclusion Special Interest Group of a national Public Health Association. Thirteen members of the Special Interest Group pooled their expertise from academic, research, and community service works as well as their lived experiences, often in multiple and intersecting areas of diversity, including, but not limited to: social determinants of health, gender and/or sexuality diversity, migrants and refugees, rural health, socio-economic disadvantages, disability, chronic health conditions, children and young people, elderly people, and ethnic minorities. Some of the authors have come from and/or worked in other countries, and they enriched this paper with their international perspectives. The thirteen authors ran a series of online brainstorming sessions to share and reflect on their experiences in light of the literature. The brainstorm results were discussed in several iterations and formulated into the 12 tips in this paper. During the iterative process, the authors identified 5 common concepts which underpin the 12 tips: diversity, equity, inclusion, intersectionality, and a strength-based approach (Figure 1). These five underpinning concepts acknowledge multidimensional aspects of identity that are key to inclusivity and account for historical, structural, and cultural factors.

This article aims to guide healthcare workers (HCWs) and health service managers in implementing inclusive practices that will potentially reduce health inequities. These tips are not intended to be an all-encompassing guide to inclusive practice; rather, they are an introductory guide to inform practical steps and professional education needs. Given that the topic of diversity and its intersections is broad, this article focuses on basic principles. Practical examples are provided for each principle, which should not be seen as limiting their scope and application.

For clarity, this article uses the following meanings for the key terms. Health inequity refers to systematic differences in health that are judged to be avoidable by reasonable action [5]. This goes beyond equality in recognizing that not all people have the same access to or experiences with healthcare. Diversity in this article is seen as a multi-axial, intersecting construct, as reflected in the joint expertise of the author team. It is worth noting that aspects of diversity are not static. Inclusion, as the base for inclusive practice, involves the actions and events that create the conditions necessary for populations to meet and go beyond their basic requirements in everyday living [8].

## 2. The Twelve Tips and Discussion

The 12 tips are presented below, alongside rationale, examples, and a brief discussion of each tip in order to aid in coherence and avoid repetition.

### 2.1. Tip 1: Beware of Assumptions and Stereotypes

Many diversity-related aspects of an individual, including but not limited to age, ethnicity, Indigeneity, socio-economic status, education level, health literacy, sexuality, gender identity, sex characteristics, disability, mental health, refugee or migrant background, religion/spirituality, and others that may require tailored care, might not be known at an initial appointment. Some of these characteristics are often assumed by default, i.e., that a patient is cisgender, heterosexual, and has typical sex characteristics [9]. Visible signs such as one’s use of a wheelchair may lead to assumptions about their health, or a person accompanying a patient may be assumed to be their life partner or family member. HCWs must avoid assumptions that may hinder an accurate holistic picture of a person’s medical and personal background, since unchecked assumptions can become a barrier to seeking healthcare services and corrode the establishment of inclusive, patient-centered care. An empathetic and mainstreaming approach, such as “I need to ask you several personal questions as a routine part of the process for all patients”, followed by a brief explanation, such as “These questions will help me understand your background so I could provide you with the best care”, might help to start the service encounter on a positive note.

Accuracy and respect are needed when recording and relaying personal information in referral letters or case reports. Umbrella terms such as “Black, Indigenous, and People of Color/BIPOC” or “Lesbian, Gay, Bisexual, Transgender, Queer, Intersex, Asexual, and others/LGBTQIA+” may convey an over-generalization and erasure of an individual’s personal characteristics. These umbrella terms may also assume that the group is homogeneous, which is often not the case [10]. Specific terms would be more informative and respectful such as “Second-generation Korean-Australian” instead of Culturally and Linguistically Diverse/CALD and “cisgender lesbian” instead of LGBTQIA+. Multi-group terms such as BIPOC and LGBTQIA+ must only be used when addressing a shared identity among all the various groups represented under that term.

### 2.2. Tip 2: Replace Labels with Appropriate Terminology

In healthcare settings and in education of health professionals, identifying current and best-practice terminology is important for optimal patient-centered care, in curriculum development, and for knowledge translation. HCWs should keep abreast with best-practice terminology for patient-centered care. Patients might prefer language which is neutral, non-judgmental, based on facts or biology [11], and/or emphasizing health or health behavior [12]. Labels such as “schizophrenics” or “the obese” may equate the person to the condition, and terms such as “non-compliant” or “non-adherence” can imply blame and judgement [11]. Preferable terms include factual phrases such as “blood glucose is high” [13]; words that reflect collaborative goal setting, such as “concordance” instead of “non-compliant” [14]; and neutral descriptors such as “living with” instead of “suffering from” [15].

Guidelines may exist for preferred language and terminology, such as in the Aboriginal and Torres Strait Islander context in Australia [16]. While using preferred language demonstrates respect, language is constantly shifting and evolving. Some terms once considered derogatory (such as queer, blind, deaf, and autistic) may now be positively embraced by some, and other terms (such as hermaphrodite) are no longer acceptable. Therefore, any list of appropriate terms must not be seen as perpetually appropriate.

### 2.3. Tip 3: Use Inclusive Language

Using inclusive language is a crucial pre-requisite for inclusive service, since language has the power to marginalize and exclude people through “othering”, which is exclusionary speech or behavior towards those who are deemed as different to oneself [17]. When providing care for diverse groups, healthcare professionals must be mindful of not speaking or behaving in a way that is perceived by those who are receiving that care as othering. Formal consensus and guidelines notwithstanding, an important principle in ensuring inclusive language is respecting the power of self-identification for individuals and communities. At the heart of inclusive language is respect for all types of diversity. Terms such as “wheelchair bound” and “confined to a wheelchair” are dismissive of the Social Model of Disability [18]; better terms include “wheelchair user” or “person who uses a wheelchair”. The term “Accessibility Action Plans” is preferred over “Disability Action Plans” due to the former’s focus on the positive rather than the negative. The lived experiences of gender-diverse people can be respected by HCWs’ use of words that patients choose in their own terms, which may include gender-neutral terms such as “chestfeed” rather than “breastfeed” [19], or binary terms such as female/woman if preferred to affirm one’s gender.

A case study can be made of the common conundrum between the use of person-first language (e.g., “a person with autism”) and identity-first language (“an autistic person”). Autistic adults commonly prefer identity-first language, while parents of children with autism may prefer person-first terminology [20,21]. In autism, as well as in other conditions, proponents of both languages have equally strong rationale. Person-first language is argued to focus on a person’s active rather than passive role in the management of their condition [22]. Identity-first language may inadvertently lead to labeling, which, among certain populations with experience of health marginalization, discrimination, racism, and judgement may reinforce barriers to healthcare access and perpetuate and entrench marginalization [7]. Conversely, identity-first language has been advocated for to move away from the idea that disability is an impairment or undesirable [21]. The use of identity-first language enables the person to claim the title and be proud of their identity, promoting autonomy, agency, and choice [21]. Identity-first language also indicates that the condition is not detachable, but integral to the individual and their perception of the world [23]. The person-first language may increase stigma toward disabled people, especially children with stigmatizing disabilities [24]. Faced with this conundrum, and other cases of differing terminology preferences within a population group, HCWs are recommended to: (a) consult each person for their preference, (b) be familiar with the relevant local guidelines, and (c) acknowledge other preferences when using one language for general communications [21,25]. In writing this discussion, for example, the authors have used both types of language in different orders as an equal acknowledgment of the two preferences.

Since people may identify with and reconcile multiple identities, HCWs need to deepen their knowledge of intersectionality, which is the relationship between multiple and intersecting social identities, and how these intersections are shaped by overlapping systems of power, privilege, and oppression [26]. Overuse of medical terminology may contribute to a hierarchical power imbalance within healthcare settings. On the other hand, avoidance of certain terms such as “heart failure” [27] may make it difficult for the patient to understand and manage their health. Language should be inclusive and respectful, but not overly simplified, and time should be taken by HCWs for adequate explanation.

### 2.4. Tip 4: Ensure Inclusivity in Physical Spaces

Physical healthcare environments which accommodate a full range of diversity and address patients’ various physical, sensory, and cognitive needs has been shown to improve the patient experience [28]. The World Health Organization produced seven modules and tip sheets to help guide health facilities towards adopting more disability-inclusive practices [29]. However, inclusive design goes beyond a requirement to meet legislative or planning regulations to focus on user-centered design. Some accessibility needs (such as ramps and elevators) may have been regulated, but not all, and HCWs should address additional needs which are yet to be regulated. For example, doorways must be wide enough for people with motorized wheelchairs or mobility scooters to pass through. Furniture, gowns, and equipment (for example, blood pressure cuffs and scales) should accommodate different body sizes [12]. Color and contrast are beneficial to aid wayfinding in patients with dementia or certain neurodevelopmental conditions [30]. It is good practice to provide a quiet room for people who may need to observe religious practices or manage over-stimulation. Venues for cultural engagement, such as a yarning circle for Aboriginal Australians [31], may support patients through inclusiveness of cultural and spiritual needs.

### 2.5. Tip 5: Use Inclusive and Appropriate Signage and Symbols

The physical, printed, and online use of signs and symbols such as LGBTQIA+ Pride/Progress flag and First Nations flags may serve as welcoming signals for diverse populations [32]. The use of these signs needs to be accompanied by competency in providing inclusive care for the respective groups to avoid disappointment. The design and format of signs, symbols, and visual aids need to consider accessibility and recognition by people with diverse needs. These considerations include factors such as font size and type, the colors used, contrast, tactile elements, depiction of people (e.g., ages, ethnicities, abilities), the physical location of the signage (ideally in a well-lit position at eye level), and appropriate use of vocabulary and local terms. The clientele’s health literacy should also be considered, and never simply assumed. Certain acronyms or words may not be readily understood by service users, for example, “ENT” (Ear, Nose, and Throat) or “ambulatory care” [33]. Alternative signage and communication formats would welcome and support people with diverse communication needs [32].

### 2.6. Tip 6: Ensure Appropriate Communication Methods

Effective communication is based on mutual respect, and often individualized by the environment, topic, and personal experiences. Therefore, in addition to using appropriate and inclusive language, HCWs should assess their clients/patients’ preferred communication methods. Effective communication in healthcare settings is based on two principles: not making assumptions and asking patients’ preferences [34]. Overlooking or dismissing patients’ preferences may erode the foundations of patient-centered care. For example, some autistic individuals may find body language and auditory input difficult to process, and prefer text-based interactions [35]. To allay fear of stigma and exclusion, these supports need to be positively framed by mainstreaming them as part of standard practice. Appropriate and preferred methods of communication, including correspondence, should be discussed with health professional students and new staff as part of their onboarding.

Culturally responsive communication is a cornerstone of person-centered care, and is inclusive of linguistic, cultural, racial, ethnic, religious, and sexual diversity [6]. An appreciation of the background of patients is essential to prevent frustrating and ineffective consultations for both HCWs and patients. Their heterogeneity notwithstanding, migrants and ethnic minorities often experience significant health disparities as well as barriers to navigating and utilizing mainstream health services effectively [36]. These individuals may be unaware of the help available or reluctant to seek help due to cultural norms [37]. Language barriers may also hinder their searching for and understanding of health information, may make it difficult to find culturally appropriate healthcare services [36]. Therefore, healthcare facilities should utilize multi-language support tools when caring for these clients/patients. HCW training in cross-cultural communication may increase the uptake of support services such as language-/culturally-specific written information, audio channels for those with low literacy in their mother tongue, cultural/language-specific training for staff members, and publicly funded interpreting services. Trained healthcare interpreters, rather than family members or friends of patients, must always be used when seeing clients/patients who are not proficient in the mainstream language [38].

### 2.7. Tip 7: Adopt a Strength-Based Approach

One common trap for HCWs is to see patients’ personal characteristics as a barrier to good health. Patients with certain features may be stereotyped as more difficult, non-compliant, or less cooperative than others. Some of these stereotypes may be based on epidemiological risk factors, such as higher prevalence of certain medical conditions among particular groups. However, there are caveats in generalizing patients. Not every person fits the stereotype of their group(s), and it is often difficult to recognize elements of unconscious biases. Furthermore, these stereotypes often see patients through a deficit lens instead of a strength-based approach [39]. Knowing that a patient resides in a rural area may be helpful in connecting them with transport or accommodation support if required, but should not lead to an assumption of vulnerability or poor compliance [40]. The deficit lens may also surreptitiously influence HCWs when asking about patients’ Indigeneity or whether they are from a migrant or refugee background. Therefore, HCWs always need to be mindful of whether they are seeing patient characteristics using a deficit lens.

The opposite of the deficit lens is the strength-based approach, which acknowledges diverse patient contexts and recognizes patients’ resilience, knowledge, capacities, strengths, and abilities [15]. A strength-based approach positively impacts healthcare service delivery [40], as has been demonstrated for Indigenous peoples [41] and neurodiverse people [42]. For First Nations peoples, it is important to use language regarding the “strong stories”, not weighted towards telling the “bad stories” to highlight deficits [43]. Storytelling and yarning are best practice methods for relating health information to First Nations people, exemplifying a strength-based approach to communication [31].

### 2.8. Tip 8: Ensure Inclusivity in Healthcare Service Research

Inclusivity should apply to all research conducted in healthcare facilities. People who are female, CALD, and live in rural areas have been under-represented in clinical trials [44] despite well-recognized health inequalities in these groups. Such an oversight may stem from language and cultural barriers, lack of trust towards researchers and institutions, marginalization, and ethnic exclusion [45,46]. Non-inclusive research practices may generate a skewed picture of the service users’ needs and may result in their walking away from the service.

It is essential that the targeted communities and research participant groups are consulted as early as possible in designing research so that their views are accommodated. Recruiting an inclusive and representative sample helps to address diverse concerns raised by participants groups and promote uptake of evidence-based recommendations. An equity lens, where priority populations form the focus of the research, may also be useful to identify barriers and enablers to accessing healthcare and improving patient outcomes. A participatory model of research, when relevant, would increase the engagement between researchers and participants [47]. Research participation disparities may be overcome by translating participants’ information into multiple languages, conducting data collection in multiple languages via trained interpreters, and employing multiple and diverse recruitment strategies and proper community engagement initiatives [48]. Procedures for remote participation (e.g., video conference or instant messaging) may enable researchers to cast a wider net in soliciting the target groups’ input and communicating about research activities. Other tips outlined in this paper should be considered when designing questionnaires and other data collection tools.

### 2.9. Tip 9: Expand the Scope of Inclusive Healthcare Delivery

It is important to look pragmatically at the structure, types, and availability of healthcare services, and to expand their scope to be more inclusive. Some groups, such as homeless and itinerant populations, are typically unable to prioritize health-seeking and may experience extreme inequity, multi-morbidity, social exclusion, and poor health outcomes [49], requiring adaptability from health services if they are to address health inequality and be truly inclusive. People with disability often have unmet healthcare needs due to barriers to accessing health care and experiences of discrimination [50]. These populations, and other marginalized groups, may benefit from services with after-hours and walk-in clinics, inter-service communication, friendly and competent staff, and holistic and integrated healthcare that considers all aspects of an individual’s physical and psychological well-being, including their social needs [7].

Considering the flexibility and scope of health service delivery demands, HCWs, researchers, and policy writers need to do more than paying lip service to the social and structural determinants of health. Coordination across health and social care can effectively meet many complex needs. Some common recommendations for inclusive service to different groups include removal of financial barriers, cultural competence training, representation of minority/marginalized groups in the health sector, and explicit inclusion of minority/marginalized individuals in health service research [51]. Addressing issues of cost and incorporating a sliding scale for users are some ways to improve affordability and access to healthcare. Evidence suggests that group-delivered health services can also improve affordability [52].

### 2.10. Tip 10: Advocate for a More Inclusive Healthcare System

While not always considered to be a HCW’s central role, advocating for patients’ needs to external parties may be required for more inclusive care. Advocating for care coordination across health and social services is needed to meet the needs of diverse populations as they navigate a complex health system. Systemic obstacles such as eligibility criteria, lack of service integration, discrimination, referral processes, waiting times, and negotiating welfare, disability, or child support, are compounded by social and structural determinants [7]. For example, patients who have traveled over 100 km from a rural area for an appointment might need to connect with accommodation or travel support to assist with health system navigation and to ensure that pathology testing, medical imaging, and appointments are coordinated [53]. HCWs’ advocacy provides a trusted bridge to communities who are marginalized, under-rated, or under-represented [54], for whom negotiating the healthcare system and accessing services may be confusing, intimidating, and overwhelming [7]. Integrating community health and social care workers’ advocacy within primary healthcare settings would promote equity and inclusion. HCWs need to discuss with the patient what shape the advocacy should take and how much information about their identity they would like to be shared, such as their ability status, cultural background, religious beliefs and practices, and other personal features.

### 2.11. Tip 11: Self-Educate on Diversity in All Its Forms

Being inclusive to diversity is not a state or end goal; rather, it is a commitment to ongoing processes of becoming *more* inclusive to *more* diversity. The many aspects of diversity are listed at the start of Tip 1, and their intersectionality permutations create an endless list of unique human characteristics. HCWs need to continually self-educate regarding the diverse needs of clients/patients and how to address them. Continuing professional development in diversity and inclusion, widening connections with diverse populations, and listening to people’s lived experiences would contribute to this lifelong learning. HCWs or health services seeking to update or change their practice should consult and co-design with the population groups which these changes will affect [55].

Being reflective of both successes and failures is an important part of learning. Reflectivity provides HCWs the opportunity to effectively examine and acknowledge any assumptions and preconceptions they bring to clinical practice or research. Routine reflection encourages HCWs to be lifelong learners through their own and others’ experiences, providing an opportunity to improve their skills in inclusive practices. Actions that went wrong may prompt the development of better ways to accomplish the goal, and initiatives that went well help to build confidence and confirm good practices.

### 2.12. Tip 12: Build Individual and Institutional Commitments

Inclusive HCWs, managers, and practices need an inclusive environment to thrive [8]. These twelve tips should be enshrined by leadership, policies, and protocols which reflect an institutional commitment to inclusivity. Inclusivity needs to be clearly visible and authentically enacted throughout the organization. Staffing, from the frontline HCWs up to the Board of Directors, should reflect the population served by the organization [56]. Diversity and inclusion must never be siloed in departments/roles where certain groups stereotypically work. Health services should encourage and facilitate participation from less-represented groups in job applications, mentoring, internships, scholarships, and awards, with dedicated opportunities when possible. Voices from all groups should be given equitable accommodation. The adage “nothing about us, without us” should always be the guide in planning, conducting, and presenting collaborative works.

Inclusivity and respect should apply equally for clients/patients, staff at all levels and professions, visitors, and business partners. If there is a zero-tolerance policy for non-inclusive staff behaviors, the policy must be equally applied in external partnerships. Managers should not assume that their service is inclusive just because they provide services to “everyone”. Instead, they should use independent tools, services, or measures to validate inclusivity practices and take a continuous quality improvement approach. Feedback and community consultation from all service user groups need to be routinely sought, analyzed, and used for improvement.

## 3. Conclusions

Healthcare workers, managers, and health service leaders need to continually strive for more inclusive practices. The twelve tips presented in this paper serve as a starting point for this endeavor. Diversity, equity, inclusion, intersectionality, and a strength-based approach are the five underpinning concepts and drivers for these twelve tips. Adopting these tips in practice is an important step to reducing health inequities faced by marginalized populations where these arise from barriers to accessing healthcare. Inclusivity is well-aligned with patient-/person-centered care, and should be incorporated into all aspects of healthcare services and organizations.

## 4. Future Direction

As evidence for the importance of inclusive healthcare practices and research continues to grow, a broader view needs to be embraced beyond day-to-day service delivery. Education in diversity and inclusion for all health professions, both in basic training and continuing professional education, is paramount. At the national level, advocacy is needed to enshrine inclusive practices in health service accreditation schemes, as well as in national guidelines for health research.

## Figures and Tables

**Figure 1 ijerph-20-04657-f001:**
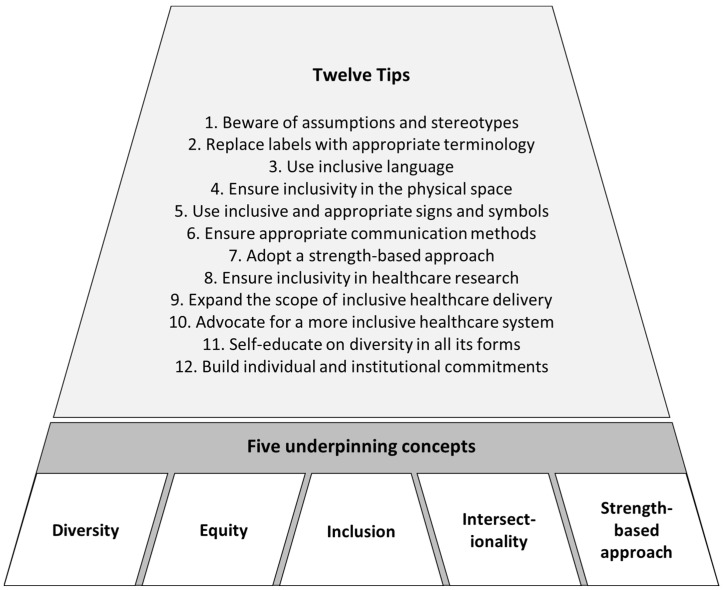
Twelve tips for inclusive practice and their five underpinning concepts.

## Data Availability

No new data were created or analyzed in this study. Data sharing is not applicable to this article.

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
