# Peer review of "Twelve Tips for Inclusive Practice in Healthcare Settings"

_ijerph, 2023, doi:10.3390/ijerph20054657_

Round 1

Reviewer 1 Report

The strength of this paper is that experts from various fields participated in the study and diverse cases from each tip were presented.

However, it is necessary to clearly present well-founded grounds for the need and the purpose of this study. Only then will the conclusion of this study and the future direction be linked.

Similar guidelines in the field of health care may be existed in each country. Nevertheless, it is necessary to discuss the importance and usefulness of the 12 tips presented in this study.

Thank you.

Reviewer 2 Report

Inequalities in health among people of various social statuses constitute one of the main challenges for public health in its meaning as a science and practical activity. Despite considerable attention to the problem since 1980, at least, striking differences in health still exist among and within countries.

Health inequality refers to the differences in the health of individuals or groups. It is the typical situation for obvious reasons. Health equity, in contrast, is a specific type of health inequality. That is systematic differences in health that could be avoided by reasonable means. Also, such differences are evaluated from moral judgement and just points of view.

One of the factors influencing people's health status is health care as such and the organisation of the system. And among them, the decisive role plays the accessibility and availability of the services. Unfortunately, many factors of different origins limit a person's access.

Health care and services of health facilities and people working there should have specific features to be evaluated as a relevant system. One of these features is an inclusive approach to the organisation and activity. As crucial as often neglected. Hence, every step to draw attention to it and show importance and possibility is valuable.

The paper "Twelve tips toward inclusive practice in healthcare settings" by Brahmaputra Marjadi and twelve distinguished persons from Australian universities, schools, and institutions have ten pages divided into a few parts: Introduction, The twelve tips and discussion, Conclusion and Future direction.

In general terms, the paper is fascinating; the issue is essential to describe it, the narration is flowing, and the tips attract attention.

 But on the other hand, the paper has substantial drawbacks that should be improved to make the document of the highest quality and eventually ready for editing.

Let's start with the critical issue but not the decisive one. When human rights are considered is better to describe The Universal Declaration of Human Rights and, after that, what is done in the WHO constitution.

I also think that in Introduction will be worthwhile to insert a few explanations of such words as equity, equality, inequities etc. It would be the real introduction to the subject. There are plenty of documents, papers, books, and reports. It should be a sort of overview about equity and inequity in health and factors impacting the availability  of services.

Next, the authors claim that " this article aims to guide healthcare workers and health service managers/ by the way managers are also healthcare workers/ to implement inclusive practices through twelve tips based in five concepts:....". I Agree. But what is the aim of this paper.?

 The best place for such a description is the first part, e.g. in the introduction. The authors should also to say a few words about "five concepts". They appear like deus ex machina and have no explanation in the following parts. Indeed, I am not thinking about vocabulary meaning but their role and importance in the study.

Also, what are the reasons for choosing them? And what is even more important is that there is no clear idea of the sources of these tips. Usually, in scientific papers, there is a part devoted to methods. A proper place could also be an Introduction. But there is nothing. Instead, there are a few words on it in the abstract. Something is better than nothing. But these few words are much too few. What was the method and procedures for selecting the tips?

Also, I don't take for granted such general information on the team as is in the abstract.

In one place, the authors insist that "this article aims to guide......." and a few words later, there is the statement that "These tips are not intended to be an all-encompassing guide.....".If so, this is a reference to the title. It should be changed to avoid discrepancies between the title and the paper's content. Add some words to the title, like "some comments"," some remarks, "and 'preliminary remarks". Otherwise, the feeling remains that the title promises more than it is in the content.

Also, the word "Perspective" above the title is not clear.

To summarize my comment and proposition regarding the first part of the paper, I think that the shape and content of the Introduction should be changed in such a way:

1.   A few sentences more of the "state of the art " referring to health inequity, influencing factors, inclusive practice problems, etc.

2.   A detailed description of the aim of the paper.

3.   A detailed description of methods and procedures of twelve tips formation.

And now, I cross over to the second part of the paper. The title is "Twelve tips and discussion.”.The general idea that the authors decided to have twelve tips and twelve discussions is strange, to some extent, but is acceptable. But the problem is in this that there is no discussion. I refer to the meaning as it is usually understood in scientific papers. In this case, there is a statement, e.g. the tip and the description, like the justification of the tip. The base for it is examples/opinions/ from the literature. Such an approach can be acceptable. But information should have its place in the text when methods are presented.

The matter of methods is also to explain why tip number twelve has any bibliographical source. It is irresistibly important. Moreover, this part's " mood " is substantially different compared with the others. It sounds like instruction or even a military chain of command.

In the end, a few words on part 3.Conclusion and 4. Future directions. And again, the such division looks strange. The future direction should be a part of the conclusion. If.? Is there a conclusion or not? That is the question.

The conclusions should emerge from the study. Indeed, when the aim of the study is unclear, conclusions are vague.

But I am willing to take my chance by saying that the aim of this study was preparing the tips as a tool for improving inclusiveness. Hence, there is only one conclusion: the twelve tips presented in this paper could serve as a starting point for it. Could serve, not serve. To check such an assumption could be the aim of the subsequent study.

Or, the authors have to change the study's aims, enlarging it by adding a new subject: and determinants of inclusiveness by reviewing the literature as a research tool.

Reviewer 3 Report

The paper embodies very clear and convincing recommendations for Health Care workers, designed to overcome the serious disadvantages in health care and outcomes experienced by the social groups listed. We are not informed about the research or personal experiences of the authors on which their recommendations are based. It would have been useful to have some indication of this. Otherwise it is a topic of great importance and publication would be valuable.
